# Scene Uyghur Recognition Based on Visual Prediction Enhancement

**DOI:** 10.3390/s23208610

**Published:** 2023-10-20

**Authors:** Yaqi Liu, Fanjie Kong, Miaomiao Xu, Wushour Silamu, Yanbing Li

**Affiliations:** 1College of Information Science and Engineering, Xinjang University, No. 777 Huarui Street, Urumqi 830017, China; 107552103674@stu.xju.edu.cn (Y.L.); 107552103730@stu.xju.edu.cn (F.K.); xmm@stu.xju.edu.cn (M.X.); wushour@126.com (W.S.); 2Xinjiang Laboratory of Multi-Language Information Technology, Xinjiang University, No. 777 Huarui Street, Urumqi 830017, China; 3Xinjiang Multilingual Information Technology Research Center, Xinjiang University, No. 777 Huarui Street, Urumqi 830017, China

**Keywords:** scene text recognition, Uyghur recognition, correction network, vision model, scene Uyghur dataset

## Abstract

Aiming at the problems of Uyghur oblique deformation, character adhesion and character similarity in scene images, this paper proposes a scene Uyghur recognition model with enhanced visual prediction. First, the content-aware correction network TPS++ is used to perform feature-level correction for skewed text. Then, ABINet is used as the basic recognition network, and the U-Net structure in the vision model is improved to aggregate horizontal features, suppress multiple activation phenomena, better describe the spatial characteristics of character positions, and alleviate the problem of character adhesion. Finally, a visual masking semantic awareness (VMSA) module is added to guide the vision model to consider the language information in the visual space by masking the corresponding visual features on the attention map to obtain more accurate visual prediction. This module can not only alleviate the correction load of the language model, but also distinguish similar characters using the language information. The effectiveness of the improved method is verified by ablation experiments, and the model is compared with common scene text recognition methods and scene Uyghur recognition methods on the self-built scene Uyghur dataset.

## 1. Introduction

Texts in natural scenes are highly generalizable and highly logical, which can intuitively convey high-level semantic information and help people analyze and understand scene content [1]. Scene Uyghur can be seen everywhere in public places in Xinjiang, such as signboards, billboards, and banners on both sides of the roads. Accurate recognition of scene Uyghur can not only provide a solid technical foundation for downstream artificial intelligence applications, such as machine translation and intelligent transportation [2], but also promote the development of information construction in Xinjiang.

In the past few years, STR [3,4] has paid more attention to the recognition of Chinese and English, and less research work has been conducted on Uyghur recognition. In recent years, thanks to the “One Belt, One Road” strategy, Uyghur recognition has developed relatively mature technologies in printing and handwriting recognition, but there are still few studies on scene Uyghur recognition.

There are two difficulties in scene Uyghur recognition. On the one hand, there is a lack of public scene Uyghur datasets in the academic community [5]. Most of the work [6,7,8] is performed using different private datasets, so the relative quality of any method cannot be compared fairly. In addition, many models [5,6] are applied to synthetic datasets, and synthetic images cannot reproduce the complexity and variability of natural images, resulting in the poor practical application of models.

Aiming at the dataset problem, this paper builds a real-scene Uyghur image dataset based on the existing dataset in the laboratory and carries out word-level annotation. The dataset contains 4000 Street View images, including road signs, billboards, and other street scenes. Some data augmentation strategies are used to expand the training set of this real scene Uyghur image dataset. In addition, this paper also constructs a Uyghur image dataset of synthetic scene by combining two text image synthesis methods, with a total of 200,000 synthetic images, to pre-train the model.

On the other hand, in addition to the common challenges faced by computer vision tasks such as background complexity and blur occlusion, scene Uyghur has its own unique challenges. In terms of morphological structure, Uyghur is a typical adhesive language [9], where characters are glued together to form concatenated segments, a feature that is particularly common in scenes. There is no uniform regulation for the width and height of characters, which are relatively random; the shape and structure vary greatly [10], which hinders the application of general character recognition algorithms. In terms of writing rules, Uyghur is written around a baseline from right to left, and many letters on the baseline have the same main body part, which can only be distinguished by subtle differences in the number and position of upper and lower punctuation marks or symbols outside the baseline. One more point and symbol or one less point and symbol will affect the shape of the character, resulting in the inability to spell or the changing of the meaning of the word [9], causing recognition errors. Figure 1 shows an example of the Uyghur features of the scene.

Aiming at visual problems such as the oblique deformation of characters, character adhesion and character similarity, this paper believes that the visual recognition ability of the model should be improved. Firstly, a set of text recognition models based on deep learning [3,4,11,12,13] were tested and analyzed on the real scene Uyghur dataset (see Section 4.6.1 for details), and ABINet [12] was identified as the basic recognition network. The first input of ABINet’s language model is visual prediction, which is corrected according to the learned language rules. However, the vision model only pays attention to visual texture information. For challenging Uyghur scenes, it is difficult to generate high-precision visual prediction. Poor-quality visual predictions increase the correction load on the language model, limiting the ability of recognition networks. Inspired by VisionLAN [13], this paper explores how to use language information in the visual space to improve the accuracy of visual predictions and improve ABINet’s vision model.

The top four probabilities output by the language model are visualized in Figure 2. The input word “
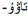
” is corrected by the language model, and the output “
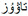
” is obtained (left side of Figure 2). However, as the number of wrong characters in the input visual prediction increases, the correction of the language model becomes challenging and may lead to wrong corrections (the input word on the right side of Figure 2 is “
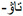
”, and the corrected word is “
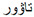
”).

The main contributions of this paper are as follows:(1)This paper constructs a real scene Uyghur image dataset and a synthetic scene Uyghur image dataset, open at https://github.com/kongfnajie/SUST-and-RUST-datasets-for-Uyghur-STR, accessed on 31 August 2023.(2)An attention-enhanced correction network TPS++ is added before the recognition network to correct the inclined Uyghur characters in the scene image into a horizontal form, which is easier to be read by subsequent recognizers.(3)The vision model of ABINet is improved. A Transformer network is introduced in U-Net to enhance the information flow along the baseline direction, aggregate horizontal features, highlight the spatial features of character positions, and alleviate the problem of character adhesion.(4)A visual masking semantic awareness module is proposed to occlude the visual features of the selected character area, and the internal relationship between local visual features is used to guide the vision model to consider the language information in the visual space to reason about the occluded characters, so as to obtain high-precision visual prediction and alleviate the correction load of the language model. When visual cues are confused, language information is used to highlight discriminative visual cues to better distinguish similar characters.

The remainder of this paper is arranged as follows. Section 2 introduces the research status of scene text recognition, text correction, and Uyghur recognition. Section 3 describes the pipeline of our model and then details each improvement in order. Section 4 shows the results of ablation experiments and comparison experiments, verifying the effectiveness of each improvement and the superiority of our method compared with other methods. Section 5 summarizes the work and offers prospects.

## 2. Related Work

### 2.1. Scene Text Recognition

It has become a trend to use language information in scene text recognition [11]. According to whether language information is used, this paper divides text recognition methods based on deep learning into two categories: language-free methods and language-aware methods.

#### 2.1.1. Language-Free Methods

Language-free methods utilize only the visual features of images for recognition. Shi et al. proposes a method based on CTC [14] using CNN to extract visual features, using RNN to model feature sequences, and finally, using CTC decoding to predict. Bai Xiang’s team applied this method to CRNN [3] for the first time, achieving recognition accuracy far beyond that of traditional methods. Most of the subsequent research has adopted the same strategy. Such models have four main stages: image preprocessing, feature extraction, sequence processing, and prediction. Along this process, the development of a pure visual recognition model is promoted by integrating multiple RNNs [15], optimizing CTC algorithm [16], guiding CTC decoding with GTC [17], and using VIT [18] to extend to SVTR [19]. Segmentation-based methods [20,21,22] treat recognition as a pixel-level classification task, which is not limited by text layout, but requires character-level annotation and is sensitive to segmentation noise. Due to language-free methods focusing solely on visual information, the recognition effect of low-quality images is poor.

#### 2.1.2. Language-Aware Methods

Language-aware methods use language information to assist recognition. The encoder-decoder architecture [23] integrates recognition cues of visual and language information, the encoder extracts image features, and the decoder predicts characters by attending to relevant information from one-dimensional image features [24] or two-dimensional image features [25]. For example, Lee et al. [26] introduces the attention mechanism to scene text recognition for the first time and learns language information by combining one-dimensional image feature sequence and character sequence embedding. Attention-based methods [27] use RNN [4] or Transformer [28,29] for language modeling. The RNN-based language model usually infers serially from left to right [30], using the characters predicted by the previous time step to model language information and identify characters one by one. This method requires expensive repetitive calculations, and the learned language knowledge is limited. Therefore, a parallel inference method [12] is introduced, which processes the sequence in parallel in the decoder and simultaneously deduces the language information of each character. Yu et al. proposes a semantic reasoning network SRN [11], which integrates visual information and semantic information and has a good recognition effect on irregular text.

The above methods also consider visual features while modeling language features. It is unknown what language features are really learned, so explicit language modeling methods are proposed [27,31,32]. ABINet [12] decouples the vision model and the language model and can pre-train the language model to obtain rich prior knowledge. The language model is an iterative bidirectional cloze network, which not only adapts to the problem of the direction of Uyghur writing from right to left, but also provides two-way language guidance for recognition. The language model iteratively corrects the prediction results and alleviates the impact of noise input, which is beneficial for scene Uyghur recognition. Therefore, ABINet is used for scene Uyghur recognition in this paper. In addition, CDistNet [31] also designs a dedicated location module to alleviate the problem of attention drift. Chu et al. [33] proposes an iterative visual modeling network, IterNet, to improve the visual prediction ability.

### 2.2. Text Correction

Part of the reason why irregular text is difficult to recognize is that the text is not presented in a standardized form, causing the text to appear distorted [34]. Popular methods usually use image preprocessing modules [35] to make the input text image easier to recognize. For example, Shi et al. [4] uses the learnable spatial transformation network STN [36] to correct irregular text to canonical form, thereby improving recognition accuracy. Luo et al. proposes a pixel-level correction network, multi-objective correction attention network MORAN [37], but the distortion correction effect of horizontal direction is poor. Recently, Zheng et al. [34] proposes a TPS++ more suitable for scene text recognition, which incorporates the attention mechanism into text correction to generate natural text corrections at the feature level that are easier to be read by subsequent recognizers. According to this method, this paper adopts special gated attention in horizontal and vertical dimensions, respectively, and finds that for Uyghur written along the horizontal baseline, horizontal attention can better locate characters, and vertical attention can better distinguish characters.

### 2.3. Uyghur Recognition

#### 2.3.1. Uyghur Printing and Handwriting Recognition

At present, Uyghur recognition has relatively mature technologies in printing [38,39,40,41,42,43,44,45,46,47,48] and handwriting [49,50,51,52]. The main research teams include Xinjiang University, Tsinghua University, Xidian University and so on.

Traditional Uyghur recognition methods [38] mostly focus on more effective character segmentation algorithms and more robust feature extraction research. Chen Qing performed heuristic feature extraction and coding for Uyghur characters, implemented a four-level classifier based on discriminant rules, and completed character recognition with template matching [39]. Bai Yunhui used the traditional recognition method of feature plus classification to realize word recognition [40]. Professor Ding Xiaoqing’s team innovated the unsegmented Uyghur recognition technology based on the implicit Markov model [10]. Lang Xiao extracts the gradient and directional line element features of Uyghur characters and uses the Euclidean distance classifier to complete the recognition task [41]. Halimulati conducts word-level recognition based on the local and overall visual features of Uyghur words and uses the N-gram language model to correct the recognition results [42].

The application of deep learning technology has promoted the development of Uyghur recognition. Li et al. first applied a CTC-based method to Uyghur recognition [43]. Li Dandan designed a “first character-word” two-stage cascade classifier [45]. Chen Yang improved the decoding order of CRNN to adapt the network to the right-to-left writing order of Uyghur [46]. The team of Professor Eskar Aimdullah used Latin forms to label Uyghur word images for text sequence recognition [47]. The TRBGA model proposed by Tang et al. realized the accurate recognition of Uyghur printing characters [48]. Zhang et al. proposed an offline signature identification method based on texture feature fusion and classified and identifies signature images by training random forests [49]. The team of Professor Maire Ibrain combined RNN and CTC to build an end-to-end online Uyghur handwriting recognition system [50].

#### 2.3.2. Scene Uyghur Recognition

Compared with printing and handwriting recognition, there is little research on scene Uyghur recognition. The outstanding work in recent years is as follows. Xiong Lijian used a spatial transformer network (TPS-STN) for preprocessing and an attention-based dense convolutional recognition network (ABDCRN) for recognition, which improves the accuracy of the self-built dataset by 11.29% compared with the baseline CRNN [6]. Fu Zilong proposes to introduce a parallel contextual attention mechanism on the parallel encoder-decoder framework to improve the visual feature alignment ability of the model, and the accuracy of the synthesized dataset is 9.00% higher than that of the baseline CRNN, but the recognition effect is poor on the samples with low contrast [5]. The team of Professor Maire Ibrain carried out text sequence recognition on Uyghur characters in images with simple backgrounds, trained them on a synthetic dataset, and evaluated them on a self-built dataset, and the accuracy was 8.20% higher than that of baseline CRNN [8]. The team of Professor Kurban Wubuli built a scene text recognition algorithm with embedded coordinate attention for multi-directional Uyghur, extracting the spatial texture information of the image, and improved the accuracy of the self-built dataset by 2.34% compared with the baseline ASTER [7]. In general, most studies use common recognition models, and these methods achieve good recognition results on their own self-built datasets. However, the recognition effect of Uyghur characters in actual scenes is not satisfactory, and there is still a large space for development in this field.

## 3. Methodology

In this section, the flow of the methods is first presented, and then each improvement is detailed in order.

### 3.1. Process

As shown in Figure 3, the proposed model is based on Transformer’s encoder–decoder framework and consists of four parts: Correction Model (CM), Vision Model (VM), Language Model (LM) and Fusion Model (FM). First, TPS++ is used to perform feature-level correction on the text in the input image, and the correction result is fed back to the two-dimensional visual features. Then, the visual features are horizontally aggregated through the improved small U-Net. In the training phase, through the Visual Masking Semantic Awareness (VMSA) module, the visual features of the selected characters are occluded to obtain a masked feature map, and the occluded characters are reasoned to obtain visual probability predictions. During the test phase, the VMSA module is removed, and the visual probability prediction is performed. Then, the visual prediction is inputted into LM for language rule learning to obtain semantic features. FM combines visual features and semantic features into fusion features for fusion probability prediction. Finally, the prediction results are input into LM for repeated iterations to correct the prediction results.

### 3.2. Correction Model

Aiming at the problem of scene Uyghur oblique deformation, the correction results of TPS [35] are not satisfactory because the calculation of TPS parameters is contentless, resulting in the corrected image only focusing on the geometric structure of the text, causing character deformation and the out-of-bounds phenomenon. This paper adopts TPS++ [34] with added attention mechanisms to suppress deformation and out-of-bounds by using text content to constrain the movement of control points.

The framework of TPS++ is shown in Figure 4, which consists of two parts: Multi-Scale Feature Aggregation (MSFA) and Attention Merging Parameter Estimation (AIPE), which are used for visual feature aggregation and attention-enhanced TPS parameter estimation, respectively.

Input image $X\in R^{W\times H\times D}$, where H and W are the height and width of the image, and D is the feature dimension. The feature maps of the first three blocks in the backbone network are input to MSFA, scaled to the same size, and concatenated from the channel dimension. Then, a codec-based feature extractor is applied for feature extraction. The shrinking path is used to extract aggregated features in shallow blocks containing more position-related cues—that is, encoding features $F_{e}\in R^{W_{e}\times H_{e}\times D}$—and the expanding path is used to extract general visual features—that is, decoding features $F_{d}\in R^{W_{d}\times H_{d}\times D}$.

AIPE uses these two features to predict the movement of control points and calculate content-based attention scores. The control points $P_{c}\in R^{W_{e}H_{e}\times 2}$ are initialized in a grid-like manner, uniformly distributed on the feature map. $F_{e}$ is used to predict the offset of each control point in the x-dimension and y-dimension, combined with the initialization coordinates, to obtain a set of regression control points $P_{c}^{`}\in R^{W_{e}H_{e}\times 2}$. The attention score matrix $A^{P\_S}\in R^{W_{d}H_{d}\times W_{e}H_{e}}$ between control points and text is adaptively computed, and the process can be formalized as follows:(1)$$A^{P\_S}=tanh(\frac{D(Fe,Fd)\times {Fe}^{⊤}}{\sqrt{D}}),$$
where $D\left(\cdot \right)$ is implemented by a dynamic gated attention block DGAB [34], and ⊤ stands for transpose.

Attention-enhanced TPS correction is performed based on control points and attention scores, and the corrected features are fed back to the backbone. TPS++ shares the visual feature extraction process with the recognition network, which saves a lot of resources compared with the correctors [4], which extract features separately in the correction stage.

### 3.3. Vision Model

VM is divided into three phases: feature extraction, sequence representation and vision prediction. The corrected features are fed back to the backbone and continue to participate in subsequent feature extraction. Then, the visual features $V_{f}\in R^{\frac{H}{4}\times \frac{W}{4}\times D}$ are obtained through a three-layer Transformer.

The position attention module generates attention maps $A^{V-S}\in R^{T\times \frac{HW}{16}}$ based on the query paradigm:(2)$$A^{V-S}=softmax(\frac{QK^{⊤}}{\sqrt{D}}),$$
(3)$$K=U(V_{f})\in R^{\frac{HW}{16}\times D},$$
where $Q\in R^{T\times D}$ is the position encoding of the character sequence, and T is the maximum length of the character sequence, which is set to 30 in this paper. $U\left(\cdot \right)$ is realized by a modified small U-Net.

Since the CM corrects the skewed Uyghur text to a horizontally represented text, the recognition model processes the characters in a horizontal arrangement. In order to adapt the recognition model to the characteristics of Uyghur adhesion, this paper introduces a Transformer network in U-Net, and the U-Net* structure is shown in Table 1. The information flow is enhanced along the baseline direction, the horizontal features are aggregated and interact through a non-local mechanism, and the spatial features of character positions are enhanced, thereby alleviating the problem of character adhesion.

Both the encoder and decoder consist of 4 convolutional layers with 3 × 3 kernels and 1 padding unit. The encoder performs downsampling by adjusting the convolution stride directly, while the decoder uses interpolation layers for upsampling. Skip connections are omitted, and functions are combined via additive operations.

LM iteratively updates semantic features and fusion features to correct visual recognition results. The initial input to the LM is the visual probability prediction $Y_{\left(0\right)}\in R^{T\times C}$, and in subsequent iterations, the input to the LM is the fusion probability prediction from the previous iteration. The process of raw visual probability prediction can be formalized as follows:(4)$$Y_{\left(0\right)}=softmax(A^{V-S}\tilde{V_{f}}W_{1}),$$
where $\tilde{V_{f}}\in R^{\frac{HW}{16}\times D}$ represents the flattened visual features, $W_{1}\in R^{D\times C}$ represents a linear transition matrix, and C represents the number of character categories.

Since VM does not participate in iterations, providing high-precision visual prediction can alleviate the correction load of LM, which has a great impact on improving the accuracy of model recognition. Therefore, this paper proposes a VMAS module, as shown in Figure 5.

In the training phase, based on the attention map $A^{V-S}$, a position P is randomly selected in the character sequence, and then the top I visual features related to the selected position are found, which are replaced by $V_{\left[mask\right]}\in R^{D}$.
(5)$$idx=G_{I}^{P}(A^{V-S}),$$
(6)$$V_{i,(i\in idx)}=V_{\left[mask\right]},$$
(7)$$V_{m}=K(V_{f},V_{i})\in R^{\frac{HW}{16}\times D},$$
where $P\in [1,N_{w}]$ represents the index of the occluded character, which is randomly obtained for the input word image with length $N_{w}$. $G_{I}(\cdot )$ is the index idx of the top I maximum value of the corresponding attention score selected according to position P on the attention map $A^{V-S}$. The values of the visual feature $V_{i}$ corresponding to these indexes are replaced by $V_{\left[mask\right]}$, and K(·) is the process of using $V_{i}$ to transform the visual feature $V_{f}$ into the masking visual feature $V_{m}$.

Remote dependencies in the visual space are modeled by stacking N transformer units. Each unit is composed of a self-attention layer and a position feedforward layer, followed by residual connection and layer normalization. The obtained visual context features $V_{c}\in R^{\frac{HW}{16}\times D}$ are sent to the prediction layer to generate visual probability prediction $Y_{\left(0\right)}\in R^{T\times C}$. The prediction process is shown in Equation (9).
(8)$$Att=\frac{{exp(M(V_{c})}_{t})}{\sum_{T}{exp(M(V_{c})}_{t})},$$
(9)$$Y_{\left(0\right)}=softmax({Att}_{t}^{T}V_{c}W_{2}),$$
where $Att\in R^{T\times 1}$ is the relation map, M(·) is the linear transformation layer, and T is the maximum length of the character sequence, which is set to 30 in this paper. ${Att}_{t}^{T}$ represents the relation score at time step t, and $W_{2}\in R^{D\times C}$ represents the linear transition matrix.

During the test phase, the VMSA module is removed, and only the prediction layer is used for recognition. Since the character information is accurately occluded during the training phase, the guided prediction layer infers the semantics of characters according to the dependencies between the visual features. Therefore, VM can adaptively consider language information in visual space and generate high-precision visual predictions.

### 3.4. Language Model

Following the method of ABINet, four Transformer decoding blocks are adopted as $F^{LM}$. It uses Q as the input and $Y_{\left(0\right)}$ as the key/value of the attention layer. By processing multiple decoder blocks, LM obtains semantic features $S\in R^{T\times D}$:(10)$$S=F^{LM}(Y_{\left(0\right)}),$$

### 3.5. Fusion Model

Conceptually, image-trained VM and text-based LM come from different patterns. Semantic features have been aligned as sequences, and to align visual context features into a sequence, a position attention module is applied again to aggregate visual context features $V_{c}$ into visual sequence features $V\in R^{T\times D}$. Visual sequence features V and semantic sequence features S are combined into fusion recognition features $F\in R^{T\times D}$ through a gating mechanism [11] for final character estimation.
(11)$$G=\sigma ([V;S]W_{3})\in R^{T\times D},$$
(12)F=G⊙V+(1−G)⊙S,
where $W_{3}\in R^{2D\times D}$ is the weight, [;] means concatenation, and ⊙ is the element-wise product. Finally, the linear layer and SoftMax function are applied on F to estimate the character sequence $Y_{\left(1\right)}\in R^{T\times C}$.

### 3.6. Loss Function

Following the approach of ABINet, supervision is added in VM, LM and FM. The loss function formula is as follows:(13)$$L=\lambda_{v}L_{v}+\frac{\lambda_{l}}{M}\sum_{i=1}^{M}L_{l}^{i}+\frac{1}{M}\sum_{i=1}^{M}L_{f}^{i},$$
where $L_{v}$, $L_{l}$ and $L_{f}$ denote the loss of vision model, language model and fusion model, respectively. M is the number of iteration rounds, set to 3. $L_{l}^{i}$ and $L_{f}^{i}$ are the losses for the ith iteration. $\lambda_{v}$ and $\lambda_{l}$ are balance factors, set $\lambda_{v}$ = $\lambda_{l}$ = 1.

The vision model loss $L_{v}$ consists of three parts: the loss $L_{m}$ for predicting masked characters, the loss $L_{d}$ for predicting other characters, and the recognition loss $L_{rec}$.
(14)$$L_{v}=\lambda_{1}L_{m}+\lambda_{2}L_{d}+L_{rec},$$
where $\lambda_{1}$ and $\lambda_{2}$ are balance factors, set $\lambda_{1}$ = $\lambda_{2}$ = 0.5.
(15)$$L_{*}=\frac{1}{T}\sum_{t=1}^{T}\log(p_{t}/g_{t}),$$

Loss $L_{*}$ uses the loss formula above, where T is the maximum length of the word, $p_{t}$ is the predicted character at time step t, and $g_{t}$ is the ground truth.

## 4. Experiments

In this section, new datasets will be constructed to train and test the proposed method, and all experiments are conducted on self-built datasets.

### 4.1. Datasets

At present, it is basically impossible to find public scene Uyghur recognition datasets. Based on the existing Uyghur recognition dataset [6] in the laboratory (Xinjiang Multilingual Information Technology Laboratory and Xinjiang Multilingual Information Technology Research Center), this paper masks effective image screening and text annotation checks to produce a real scene Uyghur image dataset. It contains 4000 Uyghur Street scene images collected in Urumqi, Kashgar and other places, of which 3200 are used as the training set, and 800 are used as test set, as shown in Figure 6. Label generation is writing word information directly into text documents.

In addition, to meet the requirements of the ABINet improvements, this paper uses SynthText [54] and the TextRecognitionDataGenerator two text image synthesis method and builds a synthetic scene Uyghur image dataset containing 200,000 composite images for the pre-training model.

The process of constructing the synthetic scene Uyghur image dataset is as follows: First, 1 million initial entries are selected from the Uyghur website. These entries undergo effective word filtering, and 500,000 entries are chosen as the foundational corpus. Then, background images are collected. This paper uses the natural scene images provided by SynthText to build the background image library. Finally, image synthesis is performed. Following the SynthText method, during the synthesis process, the maximum width-to-height ratio of text regions is set to 20, the minimum width-to-height ratio is set to 0.3, and the minimum text region area is set to 100. Each image contains no less than one Uyghur text region, resulting in the generation of 50,000 synthetic images, as shown in Figure 7a. In the synthesis process using TextRecognitionDataGenerator, the maximum word length is set to 30, the minimum length is set to 3, and random Gaussian noise, optical distortions, and other enhancements are added to enrich the training samples; this process generates 150,000 synthetic images, as shown in Figure 7b.

### 4.2. Data Augmentation

Google introduced STRAug [55], which consists of 36 image enhancement functions designed specifically for STR. Each function simulates certain image attributes found in natural scenes. These enhancement functions are categorized into eight groups: Warp, Geometry, Pattern, Blur, Camera, Process, Noise, and Weather. In this paper, five random augmentation strategies from the thirty-six are applied to 3200 real images in the training dataset. The resulting 19,200 data points serve as the training dataset, and the specific augmentation effects are illustrated in Figure 8.

### 4.3. Evaluation Criteria

This paper evaluates the recognition performance of scene Uyghur from the aspect of accuracy. The definition of accuracy is shown in the formula
(16)Acc=(ntns)×100%,
where Acc represents accuracy, $n_{t}$ represents the number of correctly recognized words, and $n_{s}$ represents the total number of sample words.

### 4.4. Experiment Settings

The model can recognize 43-character categories, including 10 numbers, 32 Uyghur characters, and 1 end-of-sequence (EOS) marker. Following ABINet [12], images are resized to 256 × 64 through data augmentation operations such as geometric transformation and perspective distortion. The feature dimension D is always set to 512, BCN is set to four layers, each layer has eight attention heads, and the number of iterations M is set to three.

In the training phase, the proportion of the number of occluded in the control batch is 10%, and the number I of visual feature masking is set to 10. Since all training images have word-level annotations, using the character index P and the original word-level annotations to generate labels for the VMSA module (for example, when P = 4 and the word is “
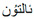
”, the labels are “
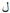
” and “
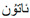
”, respectively), the label generation process is automatic without human intervention.

For different models, follow the parameter settings in their original papers. All models are trained end-to-end using the Adam optimizer, using only word-level annotations. The training batch size is 384, the initial learning rate is $10^{-4}$, the learning rate decays to $10^{-5}$ after six epochs, and the training period is set to 100 epochs. All models are trained using a server equipped with four NVIDIA AV100 GPUs.

In order to make the model converge faster, the idea of transfer learning is used in the experiment. Specifically, the synthetic dataset is used for pre-training, and then the real dataset is used for fine-tuning, during which the 10 model weights with the best training performances are saved. For the pre-training of the vision model and language model, the learning rate is kept constant during training.

### 4.5. Ablation Study

Using ABINet as the base network, several experiments are performed to prove the effectiveness of each improvement.

#### 4.5.1. Effectiveness of TPS++ Correction

The impact of common correction methods on the performance of the recognition network is studied, and the relationship between different attention methods in TPS++ and scene Uyghur recognition performance is analyzed. The number of control points is set to 4 × 16.

Baseline is ABINet [12], which does not consider correction. (W), (H), and (W + H) indicate that DGAB is used only in the horizontal dimension, DGAB is used only in the vertical dimension, and DGAB is used in both dimensions, respectively.

As shown in Table 2, ASTER uses TPS transformation correction, MORAN uses image-level grid correction, and TPS++ is applied at baseline to obtain obvious accuracy advantages, indicating that the feature-level correction integrated into the attention mechanism has a significant effect on scene Uyghur recognition effectiveness. TPS++ improves the accuracy by 0.07% when adopting DGAB in the horizontal dimension. When adopting DGAB in the vertical dimension, the accuracy is significantly improved by 0.18%. When using DGAB in both horizontal and vertical dimensions, the best accuracy is 85.55%. The reason is that Uyghur characters are arranged along the horizontal baseline, TPS++ guides the spatial position information during the feature extraction process, horizontal attention along the character direction is beneficial to better localize the characters, and the discriminative visual cues are more distributed in the vertical direction. Therefore, for scene Uyghur correction, the best results can be obtained by adopting DGAB in both dimensions.

By projecting onto the image, the corrected images of TPS and TPS++ are exemplified, as shown in Figure 9. TPS (upper part) is unable to accurately predict the control points due to the lack of guidance, and the phenomenon of character out of bounds appears. TPS++ (bottom half) uniformly initializes control points and calculates transformation parameters based on control point movement and attention scores to provide more natural text correction.

#### 4.5.2. Effectiveness of Horizontal Feature Aggregation

Due to the adhesion of Uyghur characters, the activation areas are scattered. In order to suppress the multi-activation phenomenon during the recognition process, some feature aggregation methods are explored experimentally, and it is found that enhancing the information flow along the baseline direction is of great significance for alleviating the character adhesion problem.

“Stride” is the total downsampling stride of U-Net (see Table 1). “Aggregation” is the aggregation method used in U-Net. As the downsampling operation expands the receptive field, U-Net can aggregate information. When evaluate the diverse U-Net strides’ effects on scene Uyghur recognition, as shown in Table 3(a–e), increasing the horizontal stride can bring improvement, while reducing it curtails accuracy. Furthermore, the change in horizontal stride along the baseline direction has a greater impact on the accuracy than the vertical stride. Therefore, aggregating the information along the baseline direction can improve the recognition performance of the model for scene Uyghur.

Based on this observation, this paper further explores some more general feature aggregation methods than manipulating the U-Net stride, as shown in Table 3(f–i). For U-Net with an 8 × 16 stride, operational layers are embedded after the most detailed features (1 × 8) to aggregate information. Both the mean operation and the use of convolutional layers with a kernel size of 1 × 8 can improve the performance of the model. The transformer can aggregate features along the character order through a non-local mechanism, and using a four-layer transformer can increase the accuracy by 0.19%. Although further increasing the transformer unit may achieve better results, considering efficiency and scalability, this paper finally embeds a four-layer transformer in U-Net. After aggregating horizontal features, the activation regions provide more concentrated responses, which confirms that horizontal feature aggregation can effectively suppress the multiple activation phenomenon and alleviate the problem of Uyghur character adhesion to a certain extent.

#### 4.5.3. Effectiveness of the VMSA Module

This paper proposes that the VMSA module aims to improve the visual prediction accuracy and help distinguish similar Uyghur characters. In this section, a series of ablation experiments are carried out to verify the effectiveness of VMSA.

As shown in Table 4, ABINet_V is the vision model of ABINet, using ResNet-45 and Transformer unit for STR. The first two experiments only consider visual information. Compared with the original vision model, the vision model guided by VMSA can use the language information in visual space to assist character decoding and obtain higher visual prediction accuracy. The last two experiments consider the addition of the language model. After adding the VMSA module to the baseline, the recognition accuracy increased by 0.62% compared with the baseline. It proves that the high-precision visual prediction obtained through the VMSA module can improve the correction performance of the language model.

As shown in Figure 10, similar characters (lines 1 and 2) are significantly processed by the VMSA module. For example, the character “
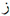
” and the character “
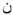
” have similar visual cues, the original vision model is wrong, while the vision model guided by VMSA gives the correct prediction of “
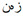
”. In addition, the correct recognition of images with complex backgrounds and poor visibility (rows 3 and 4) also proves the effectiveness of VMSA in improving the accuracy of visual prediction. For example, because the image is blurred, it is difficult to distinguish the character “
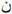
” from the character “
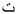
”, and the vision model guided by VMSA accurately recognizes the word “
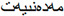
”.

#### 4.5.4. Step by Step Assessment

Improvements are made step by step to demonstrate the effectiveness of each part of the improvement.

As shown in Table 5, the TPS++, U-Net* and VMSA modules gradually enhanced the performance of the model for recognizing scene Uyghur. “√” indicates add the proposed modules. The final accuracy of the model reached 86.35%.

### 4.6. Comparative Experiments with Related Methods

#### 4.6.1. Text recognition Algorithm

Compare the performance of the method in this paper with the existing general text recognition algorithms. These mainly include RNN-based methods [3,4] and transformer-based methods [11,12,13]. For a fair comparison, this paper refers to their official code. The evaluation results are shown in Table 6.

When comparing existing text recognition methods, general text recognition methods are less effective, CRNN only uses visual features for recognition, ASTER has weak visual feature alignment ability, and language model generalization ability is not strong. ABINet shows state-of-the-art performance, achieving an 85.29% recognition accuracy and a speed of 31.5 ms per image. The visual language model ABINet is more suitable for scene Uyghur recognition, so ABINet is chosen as the baseline method. The scene Uyghur recognition method proposed in this paper outperforms other methods, benefiting from accurate visual prediction and language information-assisted character decoding, which is 1.06% higher than the baseline method. This result shows that the proposed improvements are effective. In summary, the text recognition method proposed in this paper is more suitable for scene Uyghur recognition than the general text recognition methods.

#### 4.6.2. Scene Uyghur Recognition Algorithm

Compare the performance of the method in this paper with the scene Uyghur recognition algorithms proposed in recent years. These mainly include improved methods based on CRNN [5,6,8] and improved methods based on ASTER [7]. Since there is no public code, this paper reproduces their method. For fair comparison, the settings of each model are consistent during the experiment. The experimental results are shown in Table 7.

The method in this paper achieves the best recognition performance in the comparison experiment, which reflects that the visual prediction enhancement method proposed in this paper can effectively solve the problems of oblique deformation, character adhesion and character similarity in scene Uyghur recognition. In the scene Uyghur recognition algorithms, it is very competitive.

### 4.7. Analysis of Failure Samples

This section shows samples of this paper’s method recognition failures. As can be seen from Figure 11, when encountering samples with low background contrast or severe visual interference, the model captures wrong visual information, and the recognition effect is poor. Such situations are still common difficulties in STR.

## 5. Conclusions

This paper proposes a scene Uyghur recognition model with enhanced visual prediction, which improves the basic model ABINet on the difficulty of scene Uyghur recognition. Firstly, TPS++ is used to perform feature-level correction on the text to solve the problem of slanting deformation. Then, the U-Net structure is improved to alleviate the character adhesion problem. Finally, a visual masking semantic awareness module is introduced to guide the vision model to utilize language information, which improves the accuracy of visual prediction and not only alleviates the correction load of the language model, but also facilitates the distinction of similar characters. Experiments on a publicly self-built scene Uyghur image dataset show that the model is competitive among existing scene Uyghur recognition algorithms. However, the method of horizontal feature aggregation can only alleviate the problem of character adhesion during the encoding process, and it does not fundamentally solve the problem, which needs to be improved in future work. In addition, the model structure is relatively complex, and future work will further optimize the network structure and improve the adaptability of the model based on the current research.

## Figures and Tables

**Figure 1 sensors-23-08610-f001:**
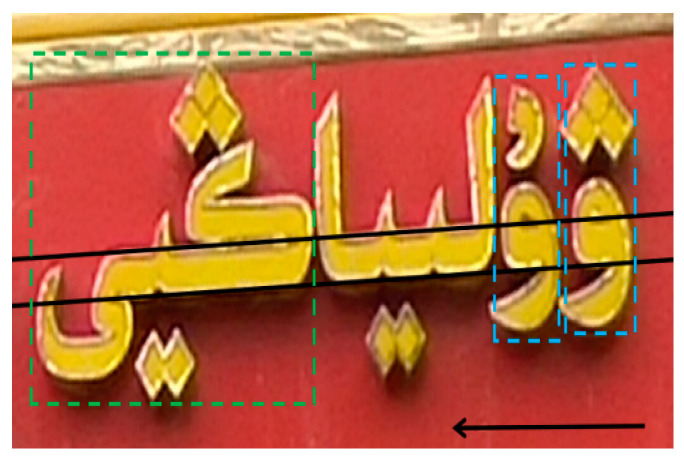
Scene Uyghur features example diagram. Black line: write along the baseline from right to left. Black arrow: writing direction. Green box: characters are glued together to form a concatenated segment. Blue box: example of similar characters.

**Figure 2 sensors-23-08610-f002:**
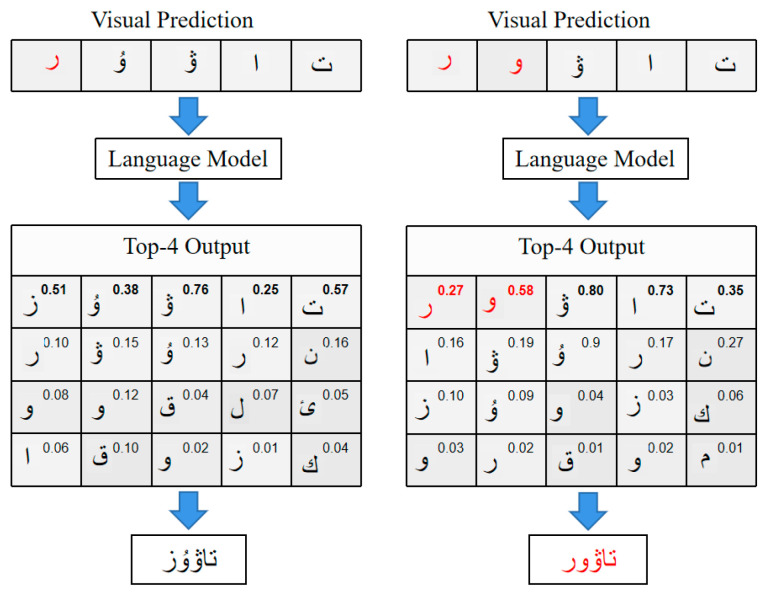
The top four recognition results of the language model. The input word is “
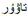
” in the left image, the input word is “
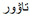
” in the right image, and the label is “
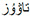
”. Numbers in the upper right corner indicate the probability of the output character, and red characters indicate incorrect predictions.

**Figure 3 sensors-23-08610-f003:**
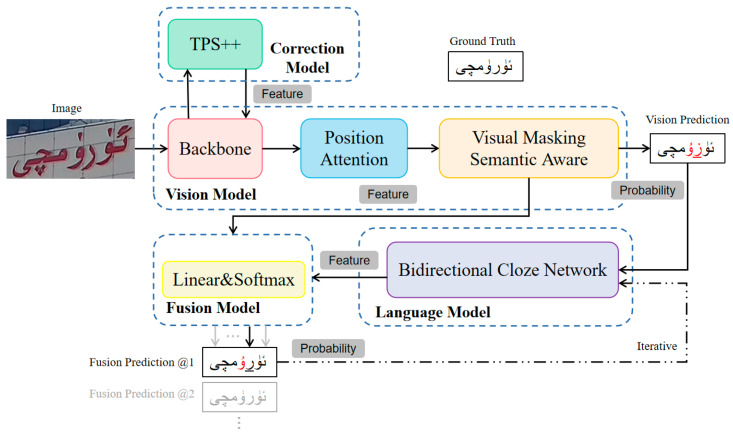
Proposed Text Recognition Model Framework. The red font is the wrong prediction.

**Figure 4 sensors-23-08610-f004:**
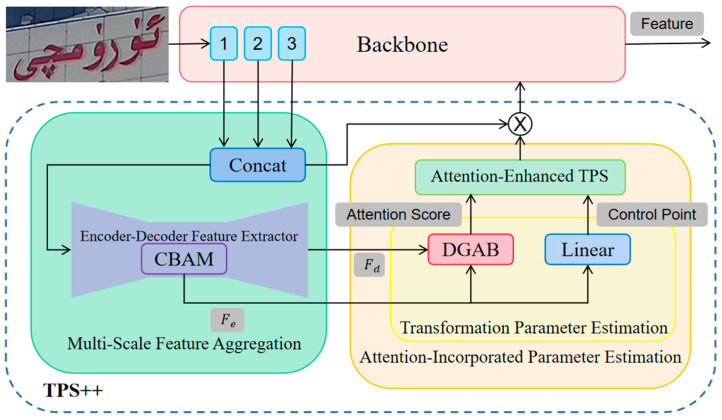
Architecture of TPS++. CBAM [53] is channel–space joint attention, which is often used to highlight important features. DGAB [34] is a specially designed dynamic gated attention block.

**Figure 5 sensors-23-08610-f005:**
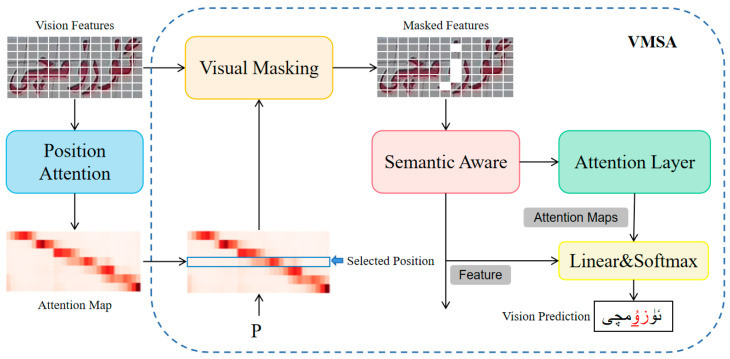
Architecture of the VMSA module. The red font is the wrong prediction.

**Figure 6 sensors-23-08610-f006:**
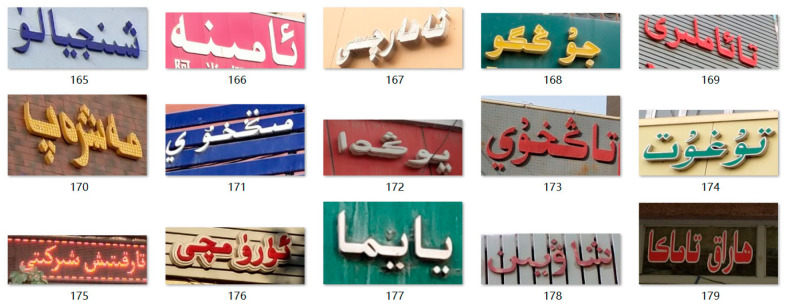
Example of real scene Uyghur recognition data.

**Figure 7 sensors-23-08610-f007:**
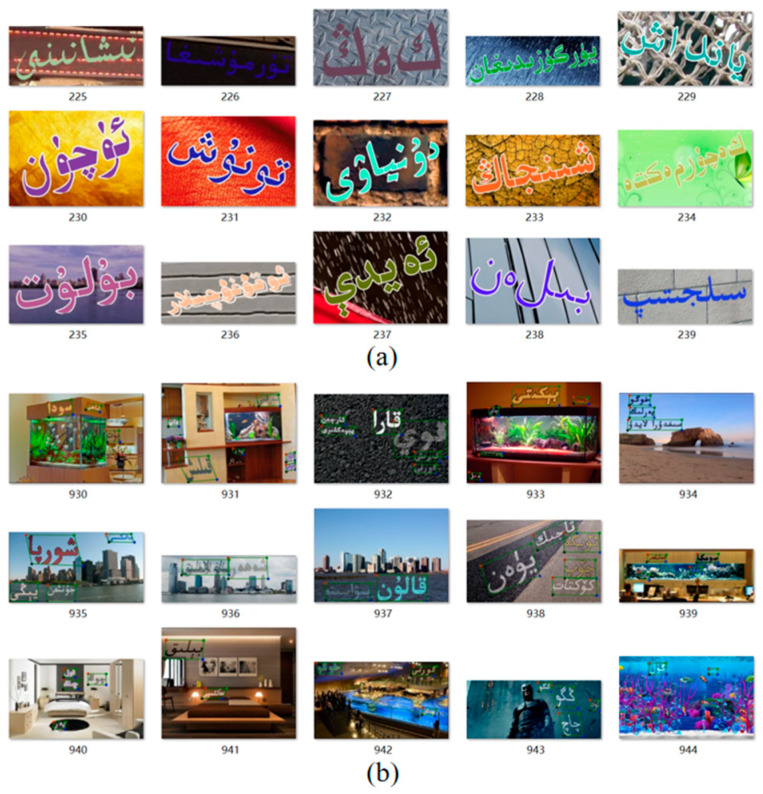
Example of synthetic scene Uyghur recognition data. (**a**) example of SynthText synthetic images. (**b**) example of TextRecognitionDataGenerator synthetic images.

**Figure 8 sensors-23-08610-f008:**
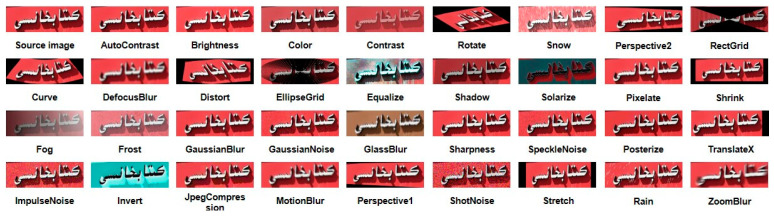
The effect of 36 augmentation strategies on the same image.

**Figure 9 sensors-23-08610-f009:**
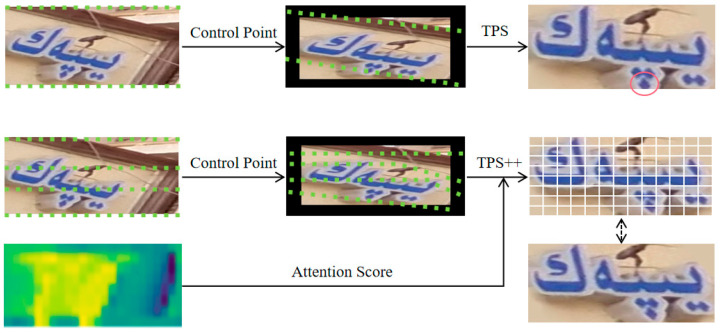
Example of correction effect comparison. Green dot: control point. Red circle: character out of bounds.

**Figure 10 sensors-23-08610-f010:**
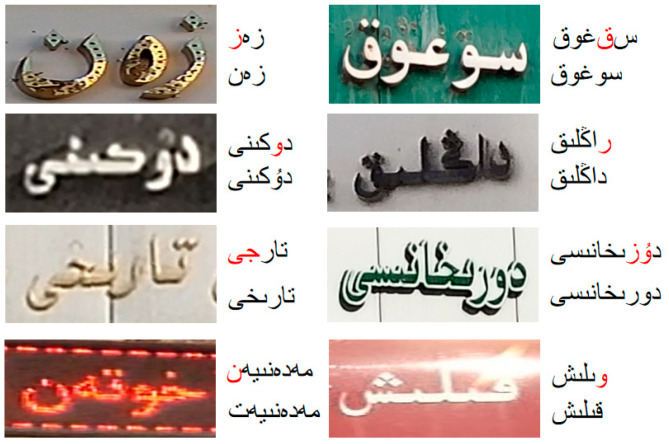
Qualitative analysis of the VMSA module. Upper string: Predictions for VMs that do not use the VMSA module. Lower String: Identification of the VM using the VMSA module. Red characters are mis predicted.

**Figure 11 sensors-23-08610-f011:**
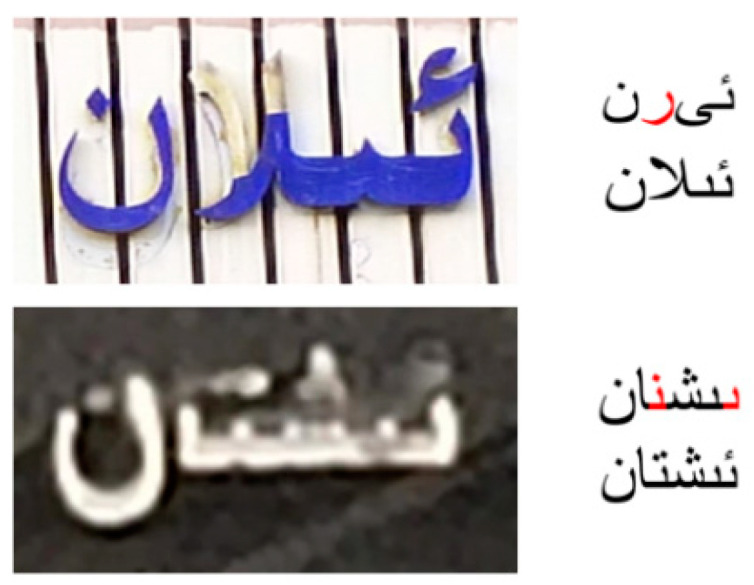
Example of failure samples. Upper string: prediction result. Lower string: ground truth label. The red parts are the wrong prediction.

**Table 1 sensors-23-08610-t001:** U-Net* structure.

Layers	Parameters (Filters, Strides)	Output (H, W)
Conv.	[64, (1,2)] × 1	(h,w/2)
Conv.	[64, (2,2)] × 1	(h/2,w/4)
Conv.	[64, (2,2)] × 1	(h/4,w/8)
Conv.	[64, (2,2)] × 1	(h/8,w/16)
Trans.	-	-
Up + Conv.	[64, (1,1)] × 1	(h/4,w/8)
Up + Conv.	[64, (1,1)] × 1	(h/2,w/4)
Up + Conv.	[64, (1,1)] × 1	(h,w/2)
Up + Conv.	[512, (1,1)] × 1	(h,w)

**Table 2 sensors-23-08610-t002:** The results of the ablation experiments for the correction model.

Methods	Attention	Accuracy
Baseline	-	85.29
ASTER [4]	-	78.76
MORAN [37]	-	75.73
TPS++	W	85.36
TPS++	H	85.47
TPS++	W + H	**85.55**

**Table 3 sensors-23-08610-t003:** Ablation experiment results with U-Net modification.

Methods	Stride	Aggregation	Accuracy
(a)	(8,16)	-	85.29
(b)	(8,128)	-	85.31
(c)	(8,1)	-	84.80
(d)	(1,128)	-	85.35
(e)	(1,1)	-	84.74
(f)	(8,16)	Mean	85.30
(g)	(8,16)	Conv-1	85.34
(h)	(8,16)	Trans-1	85.41
(i)	(8,16)	Trans-4	**85.48**

**Table 4 sensors-23-08610-t004:** Ablation experiment results of the VMSA module.

Methods	Accuracy
ABINet_V	74.58
ABINet_V + VMSA	**75.42**
ABINet	85.29
ABINet + VMSA	**85.91**

**Table 5 sensors-23-08610-t005:** Progressively improved ablation experimental results.

Baseline	TPS++	U-Net*	VMSA	Accuracy
√	-	-	-	85.29
√	√	-	-	85.55
√	√	√	-	85.76
√	√	√	√	**86.35**

**Table 6 sensors-23-08610-t006:** Text recognition model evaluation results.

Model	Accuracy	$\mathrm{Params}\;(\times 10^{6}$)	Time (ms)
CRNN [3]	72.24	7.9	6.1
ASTER [4]	78.76	21.7	73.5
SRN [11]	84.73	42.5	32.4
VisionLAN [13]	83.05	27.9	24.8
ABINet [12]	85.29	33.2	31.5
Ours	**86.35**	46.8	34.7

**Table 7 sensors-23-08610-t007:** Scene Uyghur recognition model evaluation results.

Model	Accuracy	$\mathrm{Params}\;(\times 10^{6}$)	Time (ms)
ABDCRN [6]	79.82	23.5	19.1
Fu et al. [5]	83.54	35.9	34.7
Mayire et al. [8]	78.93	32.4	31.2
ASTER + ResNet29_CA [7]	84.08	44.7	72.6
Ours	**86.35**	46.8	34.7

## Data Availability

Not applicable.

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
