# Peer review of "Scene Uyghur Recognition Based on Visual Prediction Enhancement"

_sensors, 2023, doi:10.3390/s23208610_

Round 1
Reviewer 1 Report
The paper is well organized. But some points have to be clarified.
1. There are some quantity of typos. For ex:
Line 357: "a gating mechanism [] for final character estimation"
Probably, one or more references are skipped.
2. The main trouble of the article is the research method. The authors used 500,000 synthetic images and only 4,000 Uyghur Street scene images.
There were 100 times more synthetic images than regular ones. This is a very strong imbalance between regular and synthetic images. At the same time, synthetic images have hidden internal dependencies that ordinary images may not have. Therefore, neural networks are quickly over-fitted on synthetic images.
The authors have to include the proof that so huge amount of synthetic images and so big disbalance of the synthetic and normal images is appropriate in the considered problem.
Reviewer 2 Report
The paper proposes a scene Uyghur recognition model with enhanced visual prediction, in order to improve the basic model ABINet on the difficulty of scene Uyghur recognition. A real scene Uyghur image dataset and a synthetic scene Uyghur image dataset is released.
The paper is well organized.
Minor comments:
1. Some parameters are not corrected or no expressed, such as Eq. (1).
2. What is “Baseline” in Table 2? Please give description in details.
Round 2
Reviewer 1 Report
The article has improved significantly. All my comments have been taken into account. I recommend the article for publication.